# Controlling TIME: How MNK Kinases Function to Shape Tumor Immunity

**DOI:** 10.3390/cancers12082096

**Published:** 2020-07-28

**Authors:** Thao N.D. Pham, Christina Spaulding, Hidayatullah G. Munshi

**Affiliations:** 1Department of Medicine, Feinberg School of Medicine, Northwestern University, Chicago, IL 60611, USA; christina.spaulding@northwestern.edu; 2Jesse Brown VA Medical Center, Chicago, IL 60612, USA; 3Robert H. Lurie Comprehensive Cancer Center, Chicago, IL 60611, USA

**Keywords:** MNK kinases, eIF4E, hnRNP A1, Sprouty, macrophages, neutrophils, lymphocytes, tumor immune microenvironment (TIME), immune responses

## Abstract

A number of studies have clearly established the oncogenic role for MAPK-interacting protein kinases (MNK) in human malignancies. Modulation of MNK activity affects translation of mRNAs involved in cancer development, progression, and resistance to therapies. As a result, there are ongoing efforts to develop and evaluate MNK inhibitors for cancer treatment. However, it is important to recognize that MNK activity also plays an important role in regulating the innate and adaptive immune systems. A better understanding of the role of MNK kinases and MNK-mediated signals in regulating the immune system could help mitigate undesired side effects while maximizing therapeutic efficacy of MNK inhibitors. Here, we provide a systematic review on the function of MNK kinases and their substrates in immune cells.

## 1. Introduction

### 1.1. MAPK-Interacting Serine/Threonine-Protein Kinases

The MAPK-interacting kinases 1 and 2, or MNK1 and MNK2, are serine/threonine protein kinases that were identified as substrates for MAPK/ERK and p38 MAPK by two different groups [1,2]. Phosphorylation of MNK kinases by MAPKs allows MNKs to interact with and phosphorylate a number of downstream effectors, including eukaryotic initiation factor 4E (eIF4E) [3], heterogeneous nuclear ribonucleoprotein A1 (hnRNP A1) [4], and Sprouty 1,2 [5,6]. These proteins, in turn, participate in a myriad of signals that culminate in distinct biological responses. It should be noted that MNK1 and MNK2 can exhibit substrate specificity [7], and therefore can elicit responses that are unique to each kinase. MNK kinases play important roles in cancer development and resistance to therapeutics. Targeting MNK kinases has the potential to delay tumor growth and reverse drug resistance. For example, blockade of MNK activity enhances sensitivity to BET inhibitors [8] and gemcitabine [9] in pancreatic cancer, mTOR inhibitors in leukemia [10] and lung cancer [11], and trastuzumab in breast cancer [12]. More recently, it was demonstrated in the highly metastatic Myc^Tg^;Kras^G12D^ hepatocellular carcinoma model that targeting MNK kinases can reprogram the tumor microenvironment, allowing for increased T-cell infiltration and function [13]. These studies emphasize the need for a better understanding of how MNK kinases and their effectors function to affect different components of the tumor microenvironment. Several of these MNK effectors have been implicated in regulating the immune compartment. 

#### 1.1.1. Eukaryotic Initiation Factor 4E (eIF4E)

eIF4E is the most well-studied and the best-characterized of the MNK substrates. To date, MNK kinases are the only known physiological kinases for eIF4E [3]. The protein is necessary for cap-dependent translation of a subset of mRNAs that typically have long 5′-untranslated regions (5′-UTRs), high G/C content, and complex hairpin structures [14]. The process of mRNA translation is initiated by the binding of the m7G cap structure at the 5′-terminus to the trimeric eIF4F complex. This complex consists of eIF4E, the scaffolding protein eIF4G, and the RNA helicase eIF4A [15,16]. The physical interaction between eIF4G and eIF4E, which is via a highly conserved motif, is crucial for the recruitment of other initiation factors and ribosomal proteins for protein synthesis [15,16]. In addition, eIF4G also serves as a docking site for MNK kinases to phosphorylate eIF4E, as it brings MNK kinases and eIF4E into proximity to each other [17]. As a result, there have been efforts at targeting this eIF4G:eIF4E interaction, in addition to MNK kinases, to modulate eIF4E phosphorylation and translation process. 

Deletion of MNK kinases results in impairment of eIF4E phosphorylation and eIF4E-mediated translation [3,18]. Even a modest increase in eIF4E expression was sufficient to cause cellular transformation [19], while expression of a non-phosphorylatable eIF4E mutant in mice resulted in delayed tumor development [20] and metastasis [21]. Like eIF4E, constitutively active (T332D), but not the kinase-dead (T2A2), MNK1 kinase accelerated lymphomagenesis [20]. These and other studies have identified key targets of eIF4E that are essential for cancer cell proliferation, survival, and migration, such as cyclin D1 and D3, c-Myc, MDM2, VEGF, Survivin, and Bcl-2 [22]. 

#### 1.1.2. Heterogeneous Nuclear Ribonucleoprotein A1 (hnRNP A1)

Heterogeneous nuclear ribonucleoproteins (hnRNPs) are the largest and most well-studied family of RNA-binding proteins with more than 20 members [23,24]. hnRNP A1 belongs to the subfamily A/B and is arguably the best known member among the hnRNPs [23,24]. hnRNP A1 can promote or inhibit translation of select genes by binding to AU-rich elements (AREs), for example AUGGGA(U) or UAGA(G), residing within the 3′-UTR of the transcripts [23,24]. hnRNP A1 was first identified as a MNK substrate in T lymphocytes [4]. In both in vitro and in vivo settings, hnRNP A1 is phosphorylated by MNK kinases at residues 192 and 310/311/312, which is abrogated upon selective MNK inhibition [4]. The molecular mechanisms by which hnRNP A1 supports malignant transformation include regulation of cell survival, alteration of cell cycle, invasion and metastasis, altered metabolism and stress adaptation [23,24]. To date, inhibition of hnRNP A1 expression and activity can be achieved with both natural and synthetic compounds [25,26,27,28]. Accordingly, we and other groups have shown that inhibition of hnRNP A1 synergizes with targeted anti-cancer therapies to further suppress tumor cell proliferation and survival [27,28].

#### 1.1.3. Sprouty 1/2

Sprouty proteins are evolutionary conserved regulators of the ERK/MAPK signaling pathway. They inhibit activation of ERK in response to a wide range of trophic factors [29,30]. In mammals, there are four Sprouty homologs (*Spry1-4*), with *Spry2* exhibiting highest evolutionary conservation [31]. Spry2 protein was first reported to be a MNK substrate by DaSilva and colleagues [5]. Growth factor-induced serine phosphorylation of Spry2 increased the protein stability, which is mediated by MNK1 at Ser112 and Ser121 residues [5]. Later on, it was demonstrated that stabilization and upregulation of the Spry1 protein, following activation of type I interferon (IFN) signaling, is also mediated by MNK kinases [6]. As expression of Spry1/2 is inhibitory to IFN-mediated signaling [6], MNK-mediated upregulation of Sprouty proteins might be a potential mechanism accounting for IFN-resistance in malignant cells. 

### 1.2. Cellular Heterogeneity of Tumor Immune Microenvironment

Tumor immune microenvironment (TIME) is a dynamic system and consists of different cellular and non-cellular elements (reviewed in [32]). It is becoming clear that both the innate (macrophages, myeloid derived suppressor cells, neutrophils, dendritic cells, and natural killer cells) and adaptive (T and B lymphocytes) immune systems are essential components of TIME and that their crosstalk plays a major role in dictating anti-tumor responses (reviewed in [32,33]). Each cell type has been evaluated and associated with different prognosis and therapeutic responses in patients (reviewed in [32,33]). 

#### 1.2.1. Innate Immunity

The innate immune compartment includes cell types such as macrophages, neutrophils, natural killer cells, and mast cells [33]. Among these, the antigen-presenting cells, namely macrophages and dendritic cells, can recognize, process, and present non-self antigens to lymphocytes [34]. Depending on the pathogen identity, the non-self antigens can be coupled with either major histocompatibility complex (MHC) class I or MHC class II molecules (reviewed in [34]). The engagement of MHC molecules with T-cell antigen receptors determines T-cell differentiation and function as discussed later in this review (reviewed in [35]).

##### Macrophages

Macrophages are the most abundant tumor-infiltrating leukocytes [36,37]. Depending on local cues, infiltrating macrophages can assume either an anti (M1)- or pro (M2)-tumorigenic phenotype (tumor-associated macrophages, TAMs) (reviewed in [38]). It is noteworthy that the commonly used M1/M2 terminology refers to two extreme states of macrophage polarization, and does not by any means capture the wide range of functional plasticity in macrophages. The M1 phenotype can be induced by type I cytokines like TNFα and IFNγ, while type II cytokines, such as interleukin (IL)-4, IL-10, IL-13, and transforming growth factor beta (TGFβ), can promote macrophage polarization toward the M2 state. M1 and M2 polarization can also be induced by colony-stimulating factor 2 (CSF2 or GM-CSF) and colony-stimulating factor 1 (CSF1 or M-CSF), respectively (reviewed in [39]). M2 macrophages, referred to as TAMs from hereon, inhibit anti-tumor immune responses through multiple mechanisms, most notably, through depletion of amino acids and production of immunosuppressive metabolites, such as indolamine-2,3-dioxygenase (IDO) and arginase 1 (ARG1). TAMs can also suppress T-cell function through direct engagement of T-cell inhibitory proteins (e.g., B7-1/CD80, B7-2/CD86, programmed death-ligand 1 (PD-L1) and 2 (PD-L2)) and apoptotic receptors (e.g., FasL and TRAIL). Finally, through the production of IL-10 and TGFβ, TAMs can inhibit the activation, differentiation, proliferation, and function of effector T-cells, as well as promote the expansion of suppressive T-cells (reviewed in [38]). 

Pathways that regulate infiltration, survival, and immunosuppressive activity of TAMs have been identified and characterized in preclinical models. Targeting these pathways with CSF1 receptor (CSF1R) inhibitors, PI3Kγ inhibitors, or mTOR inhibitors suppresses TAM recruitment and survival and limits tumor progression as a result (reviewed in [38]). Notably, targeting CSF1R can also re-polarize the remaining TAMs towards the M1 phenotype and increase their anti-tumorigenic activity [40,41]. 

##### Myeloid-Derived Suppressor Cells

Myeloid-derived suppressor cells (MDSCs), which express CD11b and Gr1, refer to a group of myeloid cells that suppress immune responses (reviewed in [42]). These cells are rare in healthy individuals and are only found in pathological conditions. Under physiological conditions, GM-CSF can drive myelopoiesis while M-CSF promotes the differentiation of macrophages (reviewed in [42]). In pathological conditions, such as cancer, increased expression of these factors results in increased generation of MDSCs. Although MDSCs can suppress a number of different cells of the immune system, the main targets of MDSCs are T-cells. The factors implicated in MDSC-mediated immune suppression include ARG1 and nitric oxide synthase 2 (NOS2). ARG1 and NOS2 directly interfere with T-cell proliferation, activity, and survival by metabolizing L-arginine, a major T-cell metabolite. In addition, MDSCs can promote the development of regulatory T-cells through secretion of immunosuppressive cytokines (reviewed in [42,43]).

Different tumor types display different levels of MDSC infiltration. When MDSCs are present, there is a strong association between high numbers of MDSCs and resistance to anti-cancer therapies (reviewed in [42,43]). In recent years, various preclinical studies and clinical trials targeting MDSCs have demonstrated promising results. For instance, it has been demonstrated that inhibitors to JAK/STAT or VEGFA block MDSC development, and agents such as all-trans retinoic acid (ATRA), vitamins A and D3, and IL-12 are able to modulate MDSC differentiation. Additionally, agents aimed at inhibiting MDSC recruitment to the tumors, such as anti-glycan, anti-IL-17, anti-CXCR2, anti-galectin-3, are also being studied (reviewed in [43]). 

##### Tumor-Associated Neutrophils

Neutrophils make up a substantial proportion of the immune infiltrate in many cancer types. Neutrophils are elevated in the circulation and accumulate in peripheral organs during tumor progression (reviewed in [44]). Similar to macrophages, cancer-related neutrophils can undergo a spectrum of functional plasticity in response to local cues found in the TIME. Among the cytokines that can program neutrophil polarization, TGFβ, granulocyte colony-stimulating factor (G-CSF or CSF3), and IFNβ are the best-studied molecules. While IFNβ signals to promote the anti-tumor (N1) phenotype, TGFβ and G-CSF are known to promote tumor- and metastasis-promoting (N2) phenotypes in neutrophils [44,45]. Interestingly, N1/N2 neutrophils may be associated with different stages of tumor development: at early stages the neutrophils remain at the edges of the tumor and are predominantly N1, but as the tumor progresses, the neutrophils infiltrate into the tumor and display an N2 phenotype [46]. The exact cues that turn on this phenotypic switch remains unknown. Mature neutrophils upregulate several receptors, including Toll-like receptor (TLR) 4 and CXCR2. Ligands of CXCR2 (such as CXCL1, CXCL2, CXCL5, and CXCL8) are expressed within the tumor microenvironment by cancer, stromal, and immune cells to recruit neutrophils (reviewed in [44]). Developing and mature neutrophils also rely on G-CSF for proliferation, mobilization, and function. N2 neutrophils strongly upregulate markers CCL17, ARG1, and MMP9, which support their immunosuppressive function (reviewed in [44]). 

Several studies have suggested that expression of PD-L1^+^ on neutrophils is indicative of their immunosuppressive function. Intra- and peri-tumoral neutrophils are found to have higher expression levels of PD-L1 compared to circulating neutrophils [47]. In addition, PD-L1^+^ neutrophils suppress effector T-cell functions and contribute to tumor progression. Treatment with anti-PD-L1 was found to sufficiently reinvigorate T-cells and control tumor growth. In addition, targeting CXCR2, which is critical for neutrophil recruitment, decreases neutrophil accumulation, reduces metastases, improves T-cell infiltration, and enhances the efficacy of anti-PD-1 therapy (reviewed in [44]).

#### 1.2.2. Adaptive Immunity

##### T Lymphocytes

T lymphocytes include TCRαβ T-cells, regulatory T-cells (Treg) (described further below), the less common TCRγδ T-cells, and T-cells expressing invariant receptors (reviewed in [48]). The two types of TCRαβ-bearing T-cells that have been largely focused on are the conventional CD4^+^ and CD8^+^ T-cells. Naïve T-cells can be differentiated into CD4^+^ T helper cells or CD8^+^ cytotoxic T-cells when their T-cell antigen receptor (TCR) engages with MHC class II and I molecules, respectively (reviewed in [34,48]). 

CD8^+^ T-cells directly kill infected or damaged cells by using lytic enzymes, such as perforin and granzymes (e.g., granzyme A, B). However, various mechanisms of immune evasion have been described and demonstrated. These mechanisms include, among others, secretion of immunosuppressive cytokines (IL-10, TGFβ) by immunosuppressive cells and cancer cells themselves (reviewed in [33]). These cells can also induce T-cell exhaustion via direct engagement with inhibitory receptors on T-cells, such as PD-1, LAG-3, and TIM3 to name a few (reviewed in [33]). As discussed earlier, it is now becoming clear that most, if not all, components of the innate immune system can be re-educated by the tumor to become pro-tumorigenic and T-cell suppressive. Immunotherapy, designed to overcome these inhibitory signals, can reinvigorate T-cells and enhance their cytotoxic activity against tumor cells. 

CD4^+^ T-cells can also infiltrate neoplastic tissues and have central roles in shaping pro- and anti-tumor immune responses (reviewed in [49]). Naïve CD4^+^ T-cells can differentiate into Tregs or one of the several CD4^+^ T helper subsets depending on their milieu at the time of TCR engagement to MHC class II (reviewed in [34]). In addition to their requirement for optimal CD8^+^ T-cell priming and memory responses, T helper cells are also essential for B-cell development in the germinal center [50]. T helper 1 (Th1) and T helper 2 (Th2) cells were the first two classes of CD4^+^ T helper cells to be discovered and characterized. While Th1 cells upregulate immune-stimulatory cytokines such as IL-2, IFNγ, and TNFα; Th2 cells are known to secrete immunosuppressive cytokines, such as IL-4, IL-6, and IL-10 (reviewed in [51]). Accordingly, the presence of Th1 cells and Th1 cytokines within the TIME is associated with improved prognosis in several human cancers, whereas Th2 T-cell polarization has been implicated as a poor prognosis indicator for several malignancies (reviewed in [51]). Several other distinct T helper subsets, such as Th17, Th9, and Th22, have also been identified and shown to have opposing roles in tumorigenesis, highlighting the extensive heterogeneity of T helper populations (reviewed in [51]). 

##### Regulatory T Cells (Tregs)

Tregs have an important role in regulating immune responses against both foreign and self-antigens, thereby preventing autoimmunity and preserving immune homeostasis (reviewed in [52]). Defined as a subset of CD4^+^ T-cells, Tregs express CD4 as well as CD25 (α chain of the high-affinity IL-2 receptor). High expression of CD25 allows Tregs to enhance their uptake of IL-2 and, as a result, deplete this important cytokine from effector T-cells. Expression of the transcription factor FOXP3 is another defining feature of murine Tregs, and is required to maintain their immunosuppressive features. Experimental depletion of FOXP3 in Tregs results in deadly lymphoproliferative autoimmunity, whereas ectopic expression of FOXP3 is sufficient to reprogram conventional CD4^+^ T-cells to have an anti-inflammatory Treg phenotype. Tregs suppress T-cell activity through secreting immunosuppressive cytokines, such as TGFβ, IL-10, and IL-35, and induce T-cell apoptosis via secretion of perforin and granzymes (reviewed in [52]).

##### B-cells

B-cells are another important component of the adaptive immune system. These cells produce antibodies against foreign antigens and are responsible for long-lasting immunological protection. B-cells mature in the bone marrow and are activated within the secondary lymphoid organs (e.g., spleen, lymph nodes) via their interaction with T-cells. After binding to foreign antigens, B-cells present them to helper T-cells as part of a complex with MHC II molecules [50]. This interaction enables T-cells to upregulate signaling molecules, such as CD40L, IL-2, -4, -5, that are important for B-cell proliferation and production of antibodies [50]. While being less abundant in tumors compared to T-cells [53,54], the presence and function of B-cells have been studied as an important prognostic factor in select human cancers [55,56,57]. Lately, tumor-infiltrating B-cells have gained attraction for their diverse functions in tumor immunity, for example by promoting Treg transformation, suppressing effector T-cell functions, and even by differentiating into regulatory B-cells, a new immunomodulatory cell subset. 

## 2. Regulation of Different Immune Compartments by MNK Kinases and Their Effectors

Immune cells can tightly control their gene expression at both the transcriptional and post-transcriptional levels. However, many signaling events in innate and adaptive immunity require gene expression changes that are too fast for de novo transcription or alternative pre-mRNA processing. In this case, changes in the stability or translation of pre-existing mRNAs can allow for more rapid and dynamic responses. As described earlier, MNK kinases and their substrates can modulate translation and stability of select mRNAs following appropriate cues. Here, we review how MNK-mediated signals can regulate proliferation, differentiation, and function of innate and adaptive immune cell types. 

### 2.1. Innate Immunity

#### 2.1.1. Macrophages

Compared to neutrophils and suppressive myeloid cells, studies on MNK substrates have been performed more often in macrophages. Pharmacological inhibition of MNK kinases attenuated agonist-induced production of the pro-inflammatory cytokines TNFα and IL-6 in mouse monocytes as well as in bone marrow-derived macrophages [58]. In addition, the treated cells were found to upregulate IL-10, a potent immunosuppressive molecule [58]. These early results suggest an important role of MNK kinases in regulating pro-inflammatory responses by macrophages. Subsequent studies have provided more evidence to support a role of MNK kinases and their substrates in the survival and function of macrophages (Figure 1). 

##### eIF4E

Expression levels of phosphorylated eIF4E usually increase in response to viral infection [59]. TLR agonists, such as lipopolysaccharide (LPS), can induce MNK-mediated phosphorylation and activation of eIF4E [3]. Compared to wild type, mouse monocytes in which MNK-mediated phosphorylation of eIF4E is blocked are less efficient at translating inflammatory cytokines such as TNFα or IL-6 in response to TLR agonists [60]. These observations suggest a functional role of eIF4E in macrophage-mediated immunity. 

In pathological conditions, such as cancer, macrophages can be guided to polarize into an anti-inflammatory, immunosuppressive phenotype (reviewed in [38,39]). The polarization of macrophages into cells with specific functional characteristics is tightly controlled by transcription factors. IFN regulatory factor-8 (IRF8) is an IFNγ-inducible transcription factor intimately involved in the differentiation and function of macrophages [61]. It has been shown that activation of Notch1 and TLR4 leads to enhanced translation of IRF8 via MAPK-MNK1-MNK2 and downstream phosphorylation of eIF4E. IRF8, in turn, induces transcription of M1 macrophage-associated genes, such as *Il12a*, *Il12b*, and *Nos2* [62]. Accordingly, inhibition of MAPK (p38 and ERK1/2) or MNK1 decreases IRF8 translation and subsequent reduction of M1 genes in an eIF4E-dependent manner [62]. Simultaneously, expression of select M2-identified genes are downregulated upon phosphorylation and activation of the MNK-eIF4E pathway [62]. In addition to maintaining the M1 phenotype, IRF8 expression in macrophages also governs an anti-metastatic program. A recent study by Twum et al. demonstrated increased lung metastases in mice with IRF8-deficient macrophages [63]. Mechanistically, IRF8 deficiency in macrophages resulted in a gene expression program enriched for metastasis-associated genes [63]. Together, these studies expand our current knowledge on IRF8 and eIF4E-mediated regulation of IRF8 in the host defense against cancer.

##### hnRNP A1

Similar to eIF4E, hnRNP A1 may function to maintain the M1 and suppress the M2 phenotype in macrophages. A main target of hnRNP A1 in macrophages is osteopontin (OPN) [64]. OPN is a glycophosphoprotein with an exposed arginine-glycine-aspartate (RGD)-containing domain that is expressed by various cell types, including macrophages [65]. OPN is upregulated by tumor-infiltrating macrophages, and is associated with infiltration of these cells within tumor specimens [66]. In a murine model of glioblastoma, depletion of OPN resulted in significantly diminished percentages of CD11b^+^CD206^+^ macrophages in the spleen and CD11b^+^F4/80^+^ macrophages in the brain [67]. Further analysis revealed that OPN expression is required to maintain an M2 gene signature [67]. Compared to undifferentiated monocytes and M1 macrophages, M2 macrophages display the highest level of OPN, and depletion of OPN significantly reduces the expression of prominent M2 genes [67]. It has been demonstrated hnRNP A1 binding to the OPN promoter significantly reduced its promoter activity. Accordingly, displacement of hnRNP A1 is required for promoter activity and transcription of the *Opn* gene [64]. This study establishes *Opn* as a transcriptional target of hnRNP A1 [64] and suggests that hnRNP A1 may have an underappreciated role in mediating the M1/M2 balance in macrophages. 

hnRNP A1 is well-known for its ability to modulate mRNA stability and translation by binding to AREs, which are often found in the 3’-UTRs of mRNAs [23,24,68,69]. Some of these mRNAs, such as *Gm-csf*, have important immune-modulatory functions. Increased *Gm-csf* mRNA turnover is associated with a proviral insertion that enhances hnRNP A1 binding to its AREs [69], indicating a positive role of hnRNP A1 in regulating *Gm-csf* mRNA stability. Like OPN, GM-CSF promotes the M1, pro-inflammatory macrophage phenotype [39]. As the cytokine can be secreted by a variety of cell types, including macrophages, T-cells, cancer cells, and fibroblasts, it remains to be investigated which particular cellular subset is more likely to depend on hnRNP A1 to drive macrophage polarization. Studies evaluating relative expression of hnRNP A1 in distinct cell types will be helpful in addressing this.

##### Spry1/2

A growing body of evidence demonstrates that type I IFN can inhibit alternative macrophage activation [70,71]. For instance, loss of type I IFN signaling increases the expression of genes associated with the M2 phenotype while decreasing the expression of genes associated with M1 macrophages in *M. tuberculosis*-infected lungs [70]. As Spry1 and Spry2 proteins are downstream effectors of IFN-mediated signaling [6], it is conceivable that their activity may play a role in macrophage polarization and function. In fact, depletion of Spry2 in macrophages reduced the expression of M1 genes (*Nos2*, *Il6*). Significantly, Spry2 knockdown also promoted the expression of genes characteristic of M2 genes (*Il10*, *Arg1*, *Chil3*) [72]. While these data suggest Spry2 may have a potential role in regulating the M1/M2 polarization in macrophages, future studies are required to characterize the extent to which Spry2^null^ macrophages suppress effector T-cell function. Interestingly, in a model of allergic asthma, Spry2 deletion significantly increased the number of macrophages [73]. As M2 cells are the major effector macrophages in allergic asthma [74], it is conceivable that Spry2 deletion increased the abundance of this population. Future detailed characterization of Spry2 function in TAMs will provide insights on how to modulate macrophage polarization and function to achieve anti-tumor immunity. 

#### 2.1.2. MDSCs

##### eIF4E

The function of eIF4E in MDSCs has recently been evaluated. Using a preclinical model of melanoma, the authors found that compared to eIF4E^WT^ mice, mice that harbor MNK-resistant, eIF4E^S209A^ exhibited significantly delayed tumor growth, reduced metastasis, and increased survival [75]. Cytokine profiling revealed a pro-MDSC cytokine signature in the eIF4E^WT^ tumor, indicating a phosphorylated eIF4E-linked immunosuppression [75]. In support of the immune suppressive cytokine signature associated with phosphorylated eIF4E expression, eIF4E^WT^ melanomas showed a significant increase in MDSCs and fewer cytotoxic CD8^+^ T-cells, compared to melanomas with phosphorylation-deficient eIF4E [75]. Finally, pharmacologic inhibition of phosphorylated eIF4E reduced MDSC abundance and immunosuppressive function, and synergized with anti-PD-1 immunotherapy to delay tumor growth and metastasis [75]. These data collectively suggest that phosphorylated eIF4E expression in MDSCs is important for their survival and function. It should be noted that systemic depletion of eIF4E expression or blocking MNK-induced eIF4E phosphorylation is sufficient to block tumorigenesis [8,9,10,11,12]. This recent study hence proposes that MDSC-specific eIF4E phosphorylation may be the major determinant of tumoral immunosuppression in melanoma. It is not clear if this phenomenon remains true for other tumor types, especially in those that are more heavily infiltrated by immunosuppressive cell types other than MDSCs. 

##### hnRNP A1

Besides the known role of hnRNP A1 in regulating select cytokines [23,24], MDSC-specific hnRNP A1 expression and function are not very well-studied. Several proteins that are important for macrophage function and polarization are also relevant in MDSCs, but whether they are subject to similar regulation is not clear. For example, recent work has identified IRF8 as a previously unrecognized negative regulator of MDSCs [76]. IRF8 expression is downregulated in the tumor-induced MDSC subset. Furthermore, downregulation of IRF8, facilitated by G-CSF and GM-CSF, was found to accelerate MDSC generation and function [76]. CD11b^+^Gr-1^+^ MDSC cells that are IRF8-deficient appear to be more immunosuppressive and pro-tumorigenic [76]. This study again highlights the role of IRF8 in promoting myeloid differentiation and maturation. While it remains to be determined if IRF8 is a downstream target of MNK signaling pathway in MDSCs as in macrophages [77], it is noted that both *G-csf* and *Gm-csf* harbor multiple AREs in their 3′UTRs, which can be targeted by hnRNP A1 [69]. OPN, a transcriptional target of hnRNP A1 in macrophages [64], has been shown to promote MDSC expansion via the STAT3 pathway [78] and suppress anti-tumor immunity by promoting extramedullary myelopoiesis [79]. Future studies are needed to evaluate if and how MNK kinases mediate expression of IRF8 and OPN in MDSCs through hnRNP A1. 

#### 2.1.3. Tumor-Associated Neutrophils

##### eIF4E

Compared to other cell types, neutrophils have a significantly shorter lifespan in the bloodstream as they frequently undergo spontaneous apoptosis [80]. In inflammatory conditions, such as within the tumor microenvironment, they survive much longer. Some neutrophil physiological agonists, such as LPS, TNFα, can also delay their spontaneous apoptosis and prolong their survival [80]. It is becoming clear that MNK activity is required for this delayed apoptosis. While inhibition of MNK activity does not affect apoptosis or necrosis per se, the inhibition results in increased neutrophil cell death following treatment with agonists [81]. Furthermore, eIF4E phosphorylation promotes neutrophil accumulation and survival in mouse tumors [21,82]. In fact, neutrophils expressing mutant eIF4E^S209A^, which cannot undergo MNK-mediated phosphorylation, were found to have lower levels of anti-apoptotic proteins, such as MCL1 and BCL2 [21]. In addition, phosphorylated eIF4E-driven survival of neutrophils is required for their ability to accumulate in the tumor and promote metastasis [21,82]. 

The possibility that neutrophil differentiation is dependent on eIF4E phosphorylation has also been evaluated. In response to G-CSF, myeloid progenitor cells can quickly differentiate to neutrophils ex vivo. However, compared to wild-type, progenitor cells with phosphorylation-deficient eIF4E significantly lose their ability to differentiate [82]. While there is no direct evidence to suggest that mature neutrophils deficient in phosphorylated eIF4E are less immunosuppressive than their counterparts, the accumulation of immature neutrophils, following eIF4E inhibition, may significantly promote an immunosuppressive TIME. 

##### Spry1/2

Studies on how Spry1 and Spry2 function in neutrophils are scarce. In one study, it was demonstrated that tamoxifen-induced deletion of Spry2 in neutrophils resulted in defective activation of Src family kinases and multiple downstream pathways including ERK1/2 and Akt [83]. Understanding that the PI3K/Akt signaling pathway is critical to several neutrophil function, including chemotaxis, degranulation, and O_2_^−^ production [84], this observation suggests that Spry2 expression is essential to maintain an inflammatory phenotype in neutrophils. However, the exact mechanism underlying how Spry2 modulates neutrophil function is currently not known.

### 2.2. Adaptive Immunity

Besides an early observation that Mnk1/2 double deficiency does not affect the development of conventional αβ T-cells [73], studies that directly analyze the role of MNK kinases in lymphocytes are limited. However, there is more evidence supporting an essential role for MNK substrates, including eIF4E and Sprouty proteins in effector T-cells. While eIF4E is more important for the differentiation and proliferation of different T-cell lineages, Sprouty proteins seem to play a major role in their effector functions (Figure 2). 

#### 2.2.1. eIF4E

The T-cell chemokine RANTES/CCL5 is transcriptionally regulated by RFLAT-1 (RANTES for late-activated T lymphocytes-1) [85]. RFLAT-1 expression is in turn regulated through cap-dependent, eIF4E-mediated translation [86]. Overexpression of eIF4E increases RFLAT-1 protein, while inhibition of either MNK kinases or eIF4E phosphorylation reduces RFLAT-1 production respectively [86]. This mechanism is thought to enable T-cells to rapidly alter their abundance of RANTES in response to various pro-inflammatory factors. RANTES-deficient CD8^+^ T-cells have poor cytokine production, and also express higher amounts of inhibitory receptors consistent with more severe exhaustion [87]. As such, one can assume that inhibition of MNK activity or eIF4E phosphorylation may result in significantly reduced RANTES expression and suboptimal T-cell-mediated immune responses as a result. 

eIF4E may also play a role in T-cell differentiation. GATA-3 is a transcription factor both necessary and sufficient for T helper type 2 (T_H_2) cell differentiation [88]. In addition to TCR engagement, mTORC1-dependent increase in the synthesis of GATA-3 protein is required [89]. While the precise mechanism of GATA-3 regulation by mTORC1 has not been elucidated, it is speculated that the effects observed are mediated by a mechanism dependent on increased eIF4E phosphorylation and activity [90]. Consistent with the proposed role for eIF4E in T-cell differentiation, inhibition of eIF4E in FOXP3^−^ CD4^+^ T cells ex vivo increases FOXP3 protein, indicating that suppression of eIF4E bolsters the differentiation of FOXP3^−^ CD4^+^ T-cells into FOXP3^+^ Tregs [91]. Similarly, impairment of eIF4E expression and/or activity in Th17 CD4^+^ T-cells decreases IL-17 and increases FOXP3, which restrains Th17 cell differentiation and promotes the development of CD4^+^ T-cells expressing FOXP3. These results strongly support the relevance of translational control in CD4^+^ T-cell fate decision. 

Finally, the involvement of eIF4E in B-cell differentiation and proliferation has been explored. Follicular helper T (T_FH_) cells are a subset of CD4^+^ T-cells and are essential for the activation of B-cells in the germinal center (reviewed in [92]). These T_FH_ cells express high levels of inducible T-cell costimulator (ICOS), which is required for their initial generation and function. ICOS also has a recognized role in maintaining the T cell–B cell interactions inside the germinal center (reviewed in [92]). Notably, it has recently been reported that ICOS regulates *Il4* mRNA translation via the eIF4E pathway. Stimulation of ICOS results in increased polysome formation on *Il4* mRNA and increased IL-4 production [93]. This supply of IL-4 from T_FH_ is thought to promote B-cell activation during the course of their physical engagement [50,93]. Results from this study not only provide new insights on the contribution of ICOS to the effector function of T_FH_ but also suggest that inhibition of eIF4E phosphorylation and activity could impede the delivery of IL-4 to cognate B-cells, leading to suboptimal humoral responses.

#### 2.2.2. hnRNP A1

Involvement of the MNK-hnRNP A1 axis in T-cell lymphocytes has largely been attributed to its cytokine regulation. Expression of many cytokines is regulated at the posttranscriptional level. For example, posttranscriptional regulatory control of TNFα expression is through AREs located in the 3′-UTR of the mRNA [4]. It has been shown that TNFα expression is activated by both the MAPK/ERK and p38 MAPK pathway via the MNK- hnRNP A1 axis [4]. Blockade of either MNK expression or activity inhibits TNFα production in T-cells, whereas MNK overexpression enhances TNFα expression [4]. Mechanistically, MNK-mediated phosphorylation of hnRNP A1 relieves its binding to *TNFα*-ARE, allowing for translation and expression of the protein [4]. A large number of immune-modulatory cytokines are regulated posttranscriptionally through AREs and hence, it is conceivable that they may also be bound and regulated by hnRNP A1. Very early studies demonstrated hnRNP A1 binding to the ARE-rich regions of *Il2* and *Gm-csf* mRNAs [69,94], but the functional readouts were not investigated. 

Like macrophages, T-cells also express high levels of OPN, which increase even further following activation by the T-cell mitogen, Concanavalin A (Con A) [95]. Stimulation of OPN, also known as the early T cell-activation antigen-1 gene (Eta-1), in human T-cells results in IL-12 production in a CD40L- and IFNγ-dependent manner [96]. These findings suggest an important role of OPN in early Th1 responses. As hnRNP A1 already functions in T-cells [4], it is of interest to see if hnRNP A1 can regulate OPN in T-cells as it does in macrophages. 

#### 2.2.3. Spry1/2

The engagement between TCR and MHC complexes not only leads to T-cell activation, but also upregulates inhibitory signals that restrain excessive T-cell activation to prevent autoimmunity and maintain homeostasis. Such inhibitory mechanisms, such as CTLA-4 and PD-1, have been explored in the context of cancer to improve T-cell function and immune responses [97,98]. Efforts to define novel intracellular inhibitory molecules identified Spry proteins [99]. While loss of Spry1 does not affect CD4^+^ or CD8^+^ T-cell development, Spry1 loss enhances their effector function with increased IL-2, IFNγ, granzyme B secretion [99,100], and proliferation [101]. Mice in which Spry1 is selectively deleted in T-cells demonstrate enhanced responses to tumor vaccines and subsequently more robust tumor rejection compared to control mice [100]. Mechanistically, upon TCR engagement, both Spry1 and Spry2 are upregulated [99,101,102] and inhibit the activation of linker for activated T-cells (LAT) and phospholipase C-γ (PLC-γ) [99,101]. Through LAT and PLC-γ, Spry1 and Spry2 inhibit activation of MAPK signaling and NFκB, NFAT, and AP-1 transcription factors, thereby limiting T-cell activation and proliferation [99,101,103]. Collectively, these studies demonstrate that Spry1 and Spry2 molecules act as negative regulators of TCR signaling. In fact, in a chronic infection setting such as HIV, virus-specific T-cells display high levels of Spry2, and inhibition of Spry2 expression enhances the HIV-specific polyfunctional response independently of the PD-1 pathway [104]. 

In addition to enhancing the survival of effector CD8^+^ T-cells, it was recently demonstrated that concurrent loss of Spry1 and Spry2 promotes the generation of memory CD8^+^ T-cells with faster recall capacity and protective immunity [102]. Mechanistically, this is a result of decreased phosphorylation and expression of Akt and Foxo1, leading to increased activity of transcription factors essential for the development of memory T-cells, namely T-bet, TCF-1, and Eomesodermin. The identification of Spry1/2 as intracellular inhibitors that function to limit T-cell proliferation, survival, and memory formation, suggests that they may be attractive targets to improve tumor immunotherapy. In agreement with this notion, Spry1^-/-^ T cells were shown to have more potent cytolytic activity against lymphoma and melanoma tumor cells [100]. 

## 3. Conclusions

MNK kinases were first recognized for their role in promoting tumor progression. In cancer cells, MNK kinases mediate expression of multiple genes that promote growth, proliferation, resistance to apoptosis, and resistance to therapy [8,9,10,11,12]. Correspondingly, inhibition of MNK kinases activity can cause proliferative arrest and apoptosis in cancer cells [9,105,106]. Notably, multiple studies have also demonstrated that MNK kinases mediate expression of multiple pro-inflammatory cytokines, suggesting a potential role in immune cells. Except for neutrophils, it appears that depletion of MNK kinases, while having some major effects on the function or phenotype of immune cells, has no negative effect on their survival. 

Attempts to target MNK kinases (and their downstream substrates) led to the development and testing of several compounds, with earlier compounds failing to advance in clinical trials due to overt toxicities [107]. In recent years, new compounds, such as eFT-508, have come to light. Having passed phase 1 trials with a promising safety profile [108], it is now being evaluated in several clinical phase II trials, including microsatellite stable solid tumors. In a preclinical study with the Myc^Tg^;Kras^G12D^ hepatocellular carcinoma model, treatment of tumor-bearing mice with eFT-508 led to a significant increase in the number of cytotoxic CD8^+^ T-cells [13]. These observations raise a question about the potential roles of MNK kinases and their mediated signals in other immune compartments in addition to cancer cells. Ongoing trials evaluating eFT-508 in combination with PD-1/PD-L1 immune checkpoint inhibitors will provide insights on the extent to which targeting MNK signaling reprograms TIME. Here, we have provided a summary of what is known about how different MNK effectors function in the various immune compartments, including effector T-cells (Table 1). It is becoming clear that while all working downstream of MNK kinases, these effectors may produce varying, or even opposing effects in a given immune cell type. Advances in tumor profiling and single-cell sequencing will allow for a more accurate evaluation on the global effect of MNK inhibitors, and may even provide insights on the tumor types that are more likely to benefit from MNK inhibitor treatment. In addition to MNK inhibitors, targeting eIF4G:eIF4E has also been explored as an approach to control eIF4E phosphorylation and activity. For example, small-molecule inhibitors of this interaction have been demonstrated to balance protein synthesis and exhibit activity against cancer cells [14,109,110,111]. 

The advent of immunotherapy has significantly transformed the treatment landscape for select human cancers. As a result, efforts to combine immune-based regimens with other therapies for enhanced anti-tumor responses are also being pursued. Recent work by the Sonenberg lab demonstrates that eIF4E activity promotes the survival of pro-metastatic neutrophils and contributes to tumor metastasis [82]. Depletion of eIF4E phosphorylation by the MNK inhibitor merestinib blocked neutrophil survival and metastatic progression. In addition, they showed that neutrophils can inhibit T-cell proliferation and function. These and other data suggest that immunotherapeutic drugs, including inhibitors of CTLA-4 and PD-1/PD-L1, are particularly promising candidates to combine with MNK inhibitors. As a result, understanding how MNK kinases and their substrates function in all the different TIME compartments, including immune cells, is critical and will guide the optimization of these new combination regimens.

## Figures and Tables

**Figure 1 cancers-12-02096-f001:**
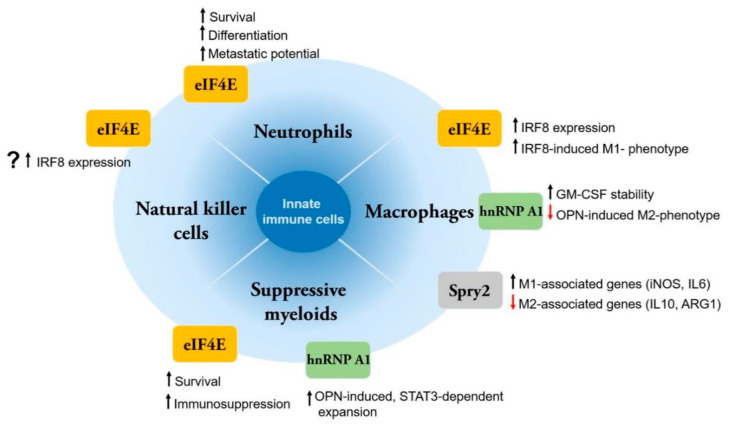
Different roles of MNK substrates in innate immune cells. Compared to neutrophils and suppressive myeloid cells, more studies on MNK substrates have been performed in macrophages. MNK substrates can modulate both survival and function of these innate cell types. Currently, there is a lack of studies in cell types such as natural killer cells or dendritic cells (not shown).

**Figure 2 cancers-12-02096-f002:**
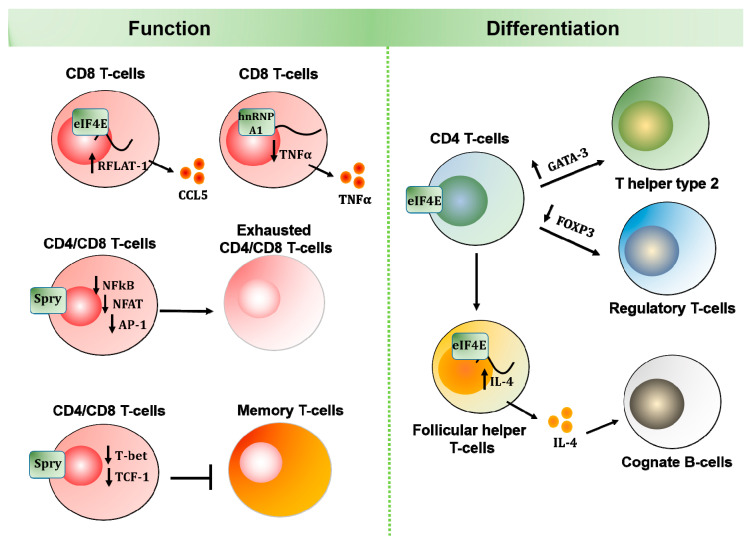
Diverse roles of MNK effectors in T-cell function and differentiation. MNK effectors have been widely studied in adaptive immune cells. While hnRNP A1 and Spry proteins play important roles in functional T-cells, eIF4E expression and activity are essential for T-cell differentiation and B-cell maturation.

**Table 1 cancers-12-02096-t001:** Summary on different roles of MNK substrates in various immune compartments.

		eIF4E	hnRNP A1	Sprouty 2
**Innate immune cells**	Macrophages	i) Promotes an M1, pro-inflammatory phenotype by regulating IRF8 [62,63] ii) Stimulates translation of pro-inflammatory cytokines TNFα, IL-6 [58]	i) Inhibits transcription and expression of OPN, which is required for M2 phenotype [64,67] ii) Increases stability of *Gm-csf* mRNAs [69,112]	i) Suppresses expression of M2-associated genes (*Il10, Arg1, Chil3*) [72] ii) Induces expression of M1-associated genes (*Nos2*, *Il6*) [72]
	Myeloid-derived suppressor cells (MDSCs)	i) May promote MDSC survival and immunosuppressive phenotype [75]	i) OPN can promote MDSC expansion via the STAT3 pathway [64,76]	
	Neutrophils	i) Promotes neutrophil survival by upregulating anti-apoptotic proteins (MCL1, BCL2) [21,82] ii) Promotes neutrophil-driven metastasis [21,82] iii) Induces myeloid-neutrophil differentiation in response to G-CSF [82]		i) Sustains activation of Src family and downstream pathways ERK1/2 and Akt [83,84]
**Adaptive immune cells**	T lymphocytes	i) Induces cap-dependent translation of RFLAT-1, which transcribes *RANTES* mRNAs [86,88] ii) Induces expression of GATA-3 for T_H_2 differentiation [88,89,90] iii) Inhibits expression of FOXP3 in CD4^+^ T-cells [90,91,113] iv) Facilitates B-cell differentiation by inducing CD4^+^ T-cell production of IL-4 [93]	i) Inhibits TNFα translation and expression by binding to *TNFα* 3′-UTR [4]	i) Inhibits expression of IL-2, IFNγ, and Granzyme B [99,100] ii) Inhibits T-cell activation and proliferation by suppressing NFκB, NFAT, and AP-1 [99,101,103] iii) Inhibits formation of memory T-cells through the activity of Akt and FoxO1 [102]

IRF8: IFN regulatory factor-8; G-CSF: granulocyte colony-stimulating factor; GM-CSF: granulocyte-macrophage colony-stimulating factor; RFLAT-1: RANTES for late-activated T lymphocytes-1; OPN: osteopontin; UTR: unstranlated region.

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
