# Peer review of "Controlling TIME: How MNK Kinases Function to Shape Tumor Immunity"

_cancers, 2020, doi:10.3390/cancers12082096_

Round 1
Reviewer 1 Report
In this review, the authors provide a general summary of MNK kinases, especially focusing on three major downstream effectors (eIF4E, hnRNAP A1 and Sprounty2) in the differentiation and functions of different immune cells in tumor immune microenvironment. The review is well-organized and overall informative. More detailed information and content about the in-depth understanding of MNK kinases in shaping tumor immunity as highlighted in the Title are needed.
- Since the title is about the MNK kinases, it is better to include additional information of the MNK kinases themselves, as well as other potential downstream targets besides the abovementioned three downstream effectors in TIME.
- In line 306 about the Spry1/2 in macrophage, could the authors provide more information on the role of Sprouty proteins in tumor immunology besides autoimmunity?
- In figure 3, could the authors include more detailed information? Like indicating the T cell subtypes in ‘Function’ and ‘Differentiation’ part, in consistence to the description in main text.
- In the conclusion, the author mentioned that some MNK kinase drugs like eFT-508 using in pre-clinical or clinical studies showed potential effects on T cells. It is really interesting to include some information or discussion on comparing the role of MNK kinases in cancer cells or immune cells in TIME.
Author Response
We thank the reviewer for their insightful feedback for our work and have edited the manuscript to address your concerns.
- Since the title is about the MNK kinases, it is better to include additional information of the MNK kinases themselves, as well as other potential downstream targets besides the abovementioned three downstream effectors in TIME.
We have now been careful to highlight all the relevant literature reporting on the potential roles of MNK kinases in TIME. Many of these studies attribute the effects to one or more of their downstream targets. Among downstream targets that are phosphorylated by MNK kinases, eIF4E, hnRNP A1, and Sprouty2 are the most well-studied and best characterized; thus, the emphasis of our manuscript on these targets. Unfortunately, other identified substrates, such as PSF.p54 nrb and cPLA2, have not been studied in the context of innate and adaptive immune systems and therefore are not included.
- In line 306 about the Spry1/2 in macrophage, could the authors provide more information on the role of Sprouty proteins in tumor immunology besides autoimmunity?
We have now modified this section to better reflect the role of Sprouty proteins in tumor immunology in addition to auto-immunity. The changes are tracked and highlighted on lines 308-312.
- In figure 3, could the authors include more detailed information? Like indicating the T cell subtypes in ‘Function’ and ‘Differentiation’ part, in consistence to the description in main text.
We have revised the indicated figure to include T-cell subtype as suggested. The revised figure (page 10) is now consistent with the description in the main text.
- In the conclusion, the author mentioned that some MNK kinase drugs like eFT-508 using in pre-clinical or clinical studies showed potential effects on T cells. It is really interesting to include some information or discussion on comparing the role of MNK kinases in cancer cells or immune cells in TIME.
We have now included a short discussion on comparing known role of MNK kinases in cancer cells versus immune cells in TIME. This new section is tracked on page 12 (lines 468-475).

Reviewer 2 Report
The authors have provided a timely review regarding the role of MNKs and its downstream signal transductors in tumor immune microenvironment. MNK1/2 kinase phosphorylates eIF4E, hnRNA A1, and Spry1/2 integrating signals from several oncogenic and immune signaling pathways (Ras, PI3K, p38, and TLRs){Wang, 1998 #89}. Phosphorylation of eIF4E selectively regulates a subset of mRNAs that mediates oncogenesis and inflammation in the prostate, skin, and breast tumor and its microenvironment. Thus, this review will attract a broad spectrum of researchers who are interested in signaling, mRNA translational control, and tumor immune-response.
However, the review suffers from many deficiencies as detailed below:
- The authors should mention that eIF4G recruits MNK kinases to phosphorylate eIF4E [PMID: 9878069]. (in the conclusion/future avenues, authors can also expand that this interaction can also be exploited for new avenues of drug discovery)
- The authors have cited the work of Ruggero et al. (ref. 12), the role of eIF4E phosphorylation in T cell infiltration and function. It would be appropriate to mention that these results were substantiated in mycTg; KrasG12D
- Line 47, it should be either, Kinase dead MNK (T2A2) or non-phosphorylatable eIF4E. The author should mention that T332D MNK (activated kinase) accelerated lymphomagenesis compared to T2A2 MNK.
- In several instances, the authors used the phrase “eIF4E is activated” (line 254, 338). The authors should either elaborate what they mean by “eIF4E is activated” or replace the term with a lucid term (eIF4E overexpression or eIF4E phosphorylation)
- p-eIF4E might enhance the translation of a subset of mRNAs, rather than general translation. This should be amended in several instances in the text.
- Line 325, it should be S209A eIF4E, instead of eIF4E deficient mice. Similarly, line 330, should be “eIF4E phosphorylation deficient”.
- Line 338, ‘unlike in macrophages where activation of eIF4E promotes an anti-tumor phenotype”. Is 67 is the correct reference for this statement?
- Line 370, eIF4E-driven survival. It should be eIF4E phosphorylation.
- Line 375, please provide a reference. Also, eIF4E phosphorylation.
- The authors should expand the conclusion by mentioning several clinical phase II trials underway for eFT508 either as a single agent or in combination.
In multiple statements, the authors confused the roles of eIF4E, overexpression, phosphorylation, or mutation.
Author Response
We thank the reviewer for your time and insightful feedback, and have edited the manuscript to address your concerns.
- The authors should mention that eIF4G recruits MNK kinases to phosphorylate eIF4E [PMID: 9878069]. (in the conclusion/future avenues, authors can also expand that this interaction can also be exploited for new avenues of drug discovery)
We have now included additional details on eIF4G and its role in MNK-mediated phosphorylation of eIF4E (page 2, lines 48-54). Under conclusion, we also mention efforts to target eIF4E/eIF4G interaction as a new therapeutic approach (page 12, lines 492-495).
- The authors have cited the work of Ruggero et al. (ref. 12), the role of eIF4E phosphorylation in T cell infiltration and function. It would be appropriate to mention that these results were substantiated in mycTg; KrasG12D.
We have now clarified that the work of Ruggero was done in MycTg;KrasG12D hepatocellular carcinoma model (page 1, lines 36-38 and page 12, line 480).
- Line 47, it should be either, Kinase dead MNK (T2A2) or non-phosphorylatable eIF4E. The author should mention that T332D MNK (activated kinase) accelerated lymphomagenesis compared to T2A2 MNK.
We have now stated that compared to constitutively active (T332D) MNK1, expression of non-phosphorylatable eIF4E and kinase-dead (T2A2) MNK1 are not sufficient to promote lymphomagenesis. This information is now included on page 2, lines 57-59.
- In several instances, the authors used the phrase “eIF4E is activated” (line 254, 338). The authors should either elaborate what they mean by “eIF4E is activated” or replace the term with a lucid term (eIF4E overexpression or eIF4E phosphorylation)
We have now clarified “eIF4E activation” term with more accurate terms, either eIF4E overexpression or phosphorylation. These changes are tracked throughout the review.
- p-eIF4E might enhance the translation of a subset of mRNAs, rather than general translation. This should be amended in several instances in the text.
The notion that eIF4E selectively enhance translation of a subset of mRNAs, rather than general translation, is now included and tracked. This is indicated on page 2, lines 44-46.
- Line 325, it should be S209A eIF4E, instead of eIF4E deficient mice. Similarly, line 330, should be “eIF4E phosphorylation deficient”.
We have now changed “eIF4E deficient mice” to “mice that harbor MNK-resistant eIF4ES209A.” This is now on lines 321-322. Similarly, we have changed “eIF4E-deficient melanomas” to “melanomas with phosphorylation-deficient eIF4E”. This is now on line 327. Both changes are tracked.
- Line 370, eIF4E-driven survival. It should be eIF4E phosphorylation.
eIF4E-driven has now been changed to phosphorylated eIF4E-driven. This is now on line 366.
- Line 375, please provide a reference. Also, eIF4E phosphorylation.
The reference has now been updated (PMID: 29463754). We have also clarified that these progenitor cells are deficient in phosphorylated eIF4E.
- The authors should expand the conclusion by mentioning several clinical phase II trials underway for eFT508 either as a single agent or in combination.
We have now expanded the conclusion by referring to ongoing clinical phase II trials including eFT-508. This is now on page 12, lines 478-480 and 484-486.
- In multiple statements, the authors confused the roles of eIF4E, overexpression, phosphorylation, or mutation.
We have carefully reviewed the manuscript, and clarified the roles of eIF4E, overexpression, phosphorylation, and mutation in studies that we referenced. These changes have been tracked in the text.

Round 2
Reviewer 2 Report
The review article can be further improved and polished, but this up to the authors